# Privacy: An Axiomatic Approach

**DOI:** 10.3390/e24050714

**Published:** 2022-05-16

**Authors:** Alexander Ziller, Tamara T. Mueller, Rickmer Braren, Daniel Rueckert, Georgios Kaissis

**Affiliations:** 1Institute of Artificial Intelligence in Medicine, Technical University of Munich, 81675 Munich, Germany; alex.ziller@tum.de (A.Z.); tamara.mueller@tum.de (T.T.M.); daniel.rueckert@tum.de (D.R.); 2Institute of Radiology, Technical University of Munich, 81675 Munich, Germany; rbraren@tum.de; 3Department of Computing, Imperial College London, London SW7 2BX, UK

**Keywords:** privacy, information flow, differential privacy, confidentiality, secrecy, privacy-enhancing technologies

## Abstract

The increasing prevalence of large-scale data collection in modern society represents a potential threat to individual privacy. Addressing this threat, for example through *privacy-enhancing technologies* (PETs), requires a rigorous definition of what exactly is being protected, that is, of privacy itself. In this work, we formulate an axiomatic definition of privacy based on quantifiable and irreducible information flows. Our definition synthesizes prior work from the domain of social science with a contemporary understanding of PETs such as differential privacy (DP). Our work highlights the fact that the inevitable difficulties of protecting privacy in practice are fundamentally information-theoretic. Moreover, it enables quantitative reasoning about PETs based on what they are protecting, thus fostering objective policy discourse about their societal implementation.

## 1. Introduction

Contemporary societies exhibit two disparate tendencies, which exist in fundamental tension. On the one hand, the tendency to collect data about individuals, processes, and phenomena on a massive scale allows robust scientific advances such as data-driven medicine, the training of artificial intelligence (AI) models with near-human capabilities in certain tasks, or the provision of a gamut of bespoke services. For fields such as medical discovery, data collection for the advancement of science can be viewed as an ethical mandate and is encouraged by regulations under the term *data altruism* [1]. On the other hand, such data collection (especially for the sole purpose of economic gain, also termed *surveillance capitalism* [2]) is critical from the perspectives of personal data protection and informational self-determination, which are legal rights in most countries. Often, the antagonism between data collection and data protection is viewed as a zero-sum game. However, a suite of technologies, termed *privacy-enhancing technologies* (PETs), encompassing techniques from the fields of cryptography, distributed computing, and information theory, promises to reconcile this tension by permitting one to draw valuable insights from data while protecting the individual. The broad implementation of PETs can thus herald a massive increase in data availability in all domains by incentivizing data altruism through the guarantee of equitable solutions to the data utilization/data protection dilemma [3].

Central to the promise of PETs is the protection of *privacy* (by design) [4]. However, the usage of this term, which is socially and historically charged and often used laxly, entails considerable ambiguity, which can hamper a rigorous and formal societal, political and legislative debate. After all, it is difficult to debate the implementation of a set of technologies when it is unclear *what exactly is being protected*. We contend that this dilemma can be resolved through a re-conceptualization of the term *privacy*. We formulated a number of expectations towards such a novel definition: it must be (1) anchored in the rich history of sociological, legal, and philosophical privacy research yet be formal and rigorous to be mathematically quantifiable; (2) easy to relate to by individuals; (3) actionable, that is, able to be implemented technologically and (4) future-proof, that is, resilient to future technological advancements, including those by malicious actors trying to undermine privacy. The key contributions of our work towards this goal can be summarized as follows:We formulated an axiomatic definition of privacy using the language of information theory;Our definition is naturally linked to *differential privacy* (DP), a PET which is widely considered the *gold standard* of privacy protection in many settings such as statistical machine learning;Lastly, our formalism exposes the fundamental challenges in actualizing privacy: Determining the origin of information flows and objectively measuring and restricting information.

## 2. Prior Work

The most relevant prior works can be distinguished into the following categories: Works by Jourard [5] or Westin [6] defined privacy as a right to restrict or control information about oneself. These definitions are relatable, as they tend to mirror the individual’s natural notion of how privacy can be realized in everyday life, such as putting curtains on one’s windows. The foundational work of Nissenbaum on *Contextual Integrity* (CI) [7] instead contends that information restriction alone is not conducive to the functioning of society. Instead, information must flow *appropriately* within a normative frame. This definition is more difficult to relate to, but it is very broad and thus suitable to capture a large number of privacy-relevant societal phenomena. Its key weakness lies in the fact that it attempts no formalization. Privacy cannot be quantified using the language of CI alone.

Our work synthesizes the aforementioned lines of thought by admitting the intuitive and relatable notion of restricting the flow of sensitive information while respecting the fact that information flow is an indispensable component of a well-functioning society.

The works of Solove [8,9,10] have followed an orthogonal approach, eschewing the attempt to define privacy directly, instead (recursively) defining it as a *solution to a privacy problem*, that is, a challenge arising during information collection, processing, or dissemination. This approach represents a natural counterpart to PETs, which represent such solutions and thus fulfill this notion of privacy. We note that, whereas our discussion focuses on DP, which is rooted in the work of Dwork et al. [11], DP is not the only PET, nor the only way to assure that our definition of privacy is fulfilled; however, the opposite holds true: DP, and every guarantee that is stronger than DP, automatically fulfills our definition presented below (provided the sender and the receiver are mutually authenticated and the channel is secure). Despite criticism regarding the guarantees and limitations of DP [12], it has established itself as the gold standard for privacy protection in fields such as statistics on large databases. We additionally discuss anonymization techniques such as *k*-anonymity [13] as examples of technologies that do not fulfill the definition of privacy we propose, as they are vulnerable to degradation in the presence of auxiliary information. For an overview of PETs, we refer to Ziller et al. [14].

Our formal framework is strongly related to Shannon’s information theory [15]. However, we also discuss a semi-quantitative relaxation of our definition, which attempts to measure qualitatively different information types (such as structural and metric information), which goes back to the work by MacKay [16].

Our work has strong parallels to the theories by Dretske [17] and Benthall [18] in that we adopt the view that the meaning and ultimately the information content of informational representations is caused by a *nomic* association with their related data.

Lastly, we note that the field of *quantitative information flow* (QIF) [19] utilizes similar abstractions as our formal framework; however, it focuses its purview more specifically to the study of *information leakage* in secure systems. It would therefore be fair to state that our framework is a generalization of QIF to a more general societal setting.

## 3. Formalism

In this section, we introduce an axiomatic framework that supports our privacy definition. All sets of numbers in this study were assumed to be nonempty and finite. We note that, while our theory has at its center abstract *entities*, one could build intuition by considering the interactions between entities as representing human communication.

**Definition** **1**(Entity)**.**
*An entity is a unique rational agent that is capable of perceiving its environment based on some inputs, interacting with its environment through some outputs and making decisions. We wrote $e_{i}$ for the ith entity in the set of entities E. Entities have a memory and can thus hold and exercise actions on some data. We wrote $d_{j}$ for the jth datum (or* item *of data) in the dataset $D^{e}$ held by e. Examples of entities include: individuals, companies, governments and their representatives, organizations, software systems and their administrators, etc.*

**Remark** **1.***The data held by entities can be owned (e.g., in a legal sense) by them or by some other entity. In acting on the data (not including sharing it with third parties), we say the entity is exercising* governance *over it. We differentiated the following forms of governance:*
*1*. Conjunct *governance: The entity is acting on its own data (i.e., data owner and governor are conjunct).**2*. Disjunct *governance: One or more entities is/are acting on another entity’s data. We distinguished two forms of disjunct governance:*Delegated *governance, where one entity is holding and/or acting on another’s data and*Distributed *governance, where >1 entity is holding parts of a single entity’s data and acting on it. Examples of distributed governance include (1) distinct entities holding and acting on disjoint subsets* (shards) *of one entity’s data (e.g., birth-date or address), (2) distinct entities holding and acting on* shares *of one entity’s data (e.g., using secret sharing schemes), and (3) distinct entities holding and acting on* copies *of one entity’s data (e.g., the IPFS protocol).*
*The processes inherent to governance are typically considered parts of the* data life-cycle *[20]. They include safekeeping, access management, quality control, deletion, etc. Permanent deletion ends the governance process.*

**Definition** **2**(Factor)**.**
*Factors are circumstances that influence an entity’s behavior. It is possible to classify factors as* extrinsic *(e.g., laws, expectations of other entities, incentives, and threats) and* intrinsic *(e.g., hopes, trust, expectations, and character), although this classification is imperfect (as there is substantive overlap) and not required for this formalism. Factors also modulate and influence each other (see the example of trust and incentives below). We wrote $f_{i}$ for the ith factor in the set of factors F.*

**Definition** **3**(Society)**.**
*A set S is called a society if and only if it contains >1 entities and ≥1 factor(s) influencing their behaviors. Our definition was intended to parallel the natural perception of a society; thus, we assumed common characteristics of societies, such as temporal and spatial co-existence. The definition is flexible insofar as it admits the isolated observation of a* useful *(in terms of modeling) subset of entities and relevant factors, such as religious or ethnic groups, which—although possibly subsets of a society in the social science interpretation—have specific and/or characteristic factors that warrant their consideration as a society. Societies undergo temporal evolution through the interaction between entities and factors. We sometimes wrote $S_{t}$ to designate a “snapshot” of society at a discrete time point t; when omitting the subscript, it is implied that we are observing a society at a single, discrete time point.*

**Definition** **4**(Communication)**.**
*Communication is the exchange of data between entities. It includes any verbal and non-verbal form of inter-entity data exchange.*

**Axiom** **1.**
*Society cannot exist without communication. Hence, communication arises naturally within society.*


**Remark** **2.**
*For a detailed treatment, compare Axiom 1 of [21].*


Our formalism is focused on a specific form of communication between entities called an *information flow*.

**Definition** **5**(Information)**.**
*Let e be an entity holding data $D^{e}$. We denote as* information *$I\subseteq D^{e}$ a structured subset of $D^{e}$ with the following properties:*
*It has a nomic association with the set of data $D^{e}$, that is, a causal relationship exists between the data and its corresponding informational representation;**The nomic association is* unique*, that is, each informational representation corresponds to exactly one datum such that the state of one item of information $X_{n}\in I$ is determined solely by the state of one datum $d_{n}\in D^{e}$;**It is measurable in the sense that* information content *is a quantitative measure of the complexity of assembling the representation of the data.*

This definition interlinks two foundational lines of work. Dretske [17] postulates that meaning is acquired through nomic association between the message’s content and the data it portrays. This aspect has been expanded upon by Benthall et al. [18], who frame nomic association in the language of Pearlian causality [22] to analyze select facets of Contextual Integrity under the lens of *Situated Information Flow Theory* [23]. The notion of information quantification as a correspondence between information content and complexity of reassembling a representation is central to information theory. We note that we utilized this term to refer to two distinct schools of thought. In the language of Shannon’s information theory [15], information content is a measure of *uncertainty reduction* about a random variable. Here, information content is measured in *Shannons* (typically synonymously referred to as *bit(s)*). Shannon’s information theory is the language of choice when discussing privacy-enhancing technologies such as DP. Our definition of information embraces this interpretation, and we will assume that—for the purposes of quantifying information—informational representations are indeed random variables. In the Shannon information theory sense, we can therefore modify our definition as follows:

**Definition** **6**(Information (in the Shannon sense))**.**
*Let e, $D^{e}$, and I be defined as above. Then, every element $X_{n}\in I$ is a random variable with mass function $p_{X_{n}}\left(x\right)$, which can be used to resolve uncertainty about a single datum $d^{n}$ through its nomic association with this datum. Moreover, the information content of $X_{n}$ is given by:*


(1)
$$I\left(X\right)=-\log_{2}p_{X_{n}}\left(x\right).$$


Moreover, our framework is also compatible with a *structural/metrical* information theory viewpoint. This perspective, which was developed alongside Shannon’s information theory and is rooted in the foundational work by MacKay [16] is a superset of the former. Here, information content in the Shannon sense is termed *selective information content* (to represent the fact each bit represents the uncertainty reduction by observing the answer to a question with two possible outcomes, i.e., selecting from two equally probable states). Moreover, information content can be *structural* (representing the number of distinguishable groups in a representation measured in *logons*) and *metrical* (representing the number of indistinguishable logical elements in a representation and measured in *metrons*). We note that the difficulty of measuring real-world information is inherent to both schools of information theory (compare also the discussion in [16]—Chapter 2).

**Definition** **7**(Information flow)**.**
*An information flow (or just flow) F is a directed transit of information between exactly two entities. We call the origin of F the* sender *S and the recipient of F the* receiver *R. The subject of F is called a* message *M and contains a single informational representation. M flows over a channel C (a medium), which we assumed to be noiseless, sufficiently capacious, and error-free. We sometimes represented a flow as:*
(2)$$F:S\overset{M}{\underset{C}{\to }}R.$$

**Remark** **3.***Flows are the irreducible unit of analysis in our framework and are****atomic****and****pairwise****. This means that they concern* exactly one datum*, and they take place between* exactly two entities*. This fact distinguishes our formalism from CI (which uses a similar terminology), where flows are defined more broadly and pertain to “communication” in a more general way, bearing strong similarities to QIF [19], where information is also viewed as flowing through a channel. Our naming for components of the flow follows standard information-theoretic literature [24].*

**Remark** **4.***We used the term* information content *of M to denote the* largest possible *quantity of information that can be derived by observing M, including the information obtained by any computation on M, irrespective of prior knowledge. This view is compatible with a* worst-case *outlook on privacy where the receiver of the message is assumed to obtain M in its entirety and make every effort available to reassemble the representation of the datum that M refers to.*

**Remark** **5.***A line of prior work, such as the work by McLuhan [25], has contended that the medium of transmission (i.e., the channel) modulates (and sometimes is a quintessential part of) the message. This point of view is not incompatible with ours, but we chose to* incorporate *the characteristics of the channel into other parts of the flow as our framework is information-theoretic but* not *communication-theoretic. For example, under our definition, an insecure (leaky) channel is regarded as giving rise to a new flow towards one or more additional receivers (see* implicit flows *below), while a corruption of the message by noise or encoding errors is deemed as directly reducing its information content. Therefore, we implicitly assumed that the state of a message is determined solely by the corresponding information that is being transmitted.*

Although flows are atomic, human communication is not: very few acts of communication result in the transmission of information only about a single datum. We thus required a tool to “bundle” all atomic flows that arise in a certain circumstance (e.g., in a certain social situation, about a specific topic, etc.). We call these groupings of flows *information flow contexts*. Moreover, communication also often happens between more than two entities (one-to-many or many-to-one scenarios). Such scenarios are discussed below.

**Definition** **8**(Information flow context)**.**
*Let $S_{t}$ be a society at time t such that $S,R\in S_{t}$, and $F_{1},\ldots ,F_{n}$ be flows $S\overset{M_{1},\ldots ,M_{n}}{\underset{C}{\to }}R$. Then, we term the collection $C_{t}=\left(S,R,F_{1},\ldots ,F_{n}\right)$ an information flow context (or just context).*

Flows are stochastic processes. This means they can *arise* randomly. The probability of their occurrence in a given society depends on numerous latent factors. Depending on the causal relationship between the appearance of a flow and an entity’s decision, we distinguished the following cases:

**Definition** **9**(Explicit flow)**.**
*An explicit flow arises as a causal outcome of a decision by the entity whose data is subject to the flow.*

**Definition** **10**(Decision)**.**
*Let e be an entity and $f=(f_{1},\ldots ,f_{n})$ a collection of factors influencing its behavior. We modeled the decision process as a random variable o∼Ber(p∣f) conditioned on the factors. Then, the* decision *$\chi_{e}$ takes the following values:*
(3)$$\chi_{e}=\{\begin{array}{cc} F & o=1\\ ⊥ & otherwise \end{array}$$

Note that Ber denotes the Bernoulli distribution, and ⊥ implies that no action is undertaken. We hypothesized the probability of decisions resulting in explicit flows to be heavily influenced by two factors. Of these, the most important is probably *trust*. In interpersonal relationships characterized by high levels of trust, entities are more likely to engage in information flows. Moreover, the reason for most societal information flows can be ultimately distilled to trust between entities on the basis of some generally accepted norm. For example, information flows from an individual acting as a witness in court towards the judge are ultimately linked to the trust in the socially accepted public order. Low levels of trust thus decrease the overall probability of an explicit flow arising. We also contend that trust acts as a barrier imposing an *upper bound* on the amount of information (described below) that an entity is willing to accept in a flow. The other main factor influencing the probability of explicit flows arising are likely *incentives*. For instance, the incentive of a larger social circle can entice individuals into engaging in explicit flows over social networks. The incentive of a free service provided over the internet increases the probability that the individual will share personal information (e.g., allow cookies). We note that—like all societal factors—incentives and trust modulate each other. In some cases, strong incentives can decrease the trust threshold required to engage in a flow, while, in others, no incentives are sufficient to outweigh trust. In addition, society itself can impose certain bounds on the incentives that are allowed to be offered or whether explicit flows are permitted *despite* high trust (e.g., generally disallowing the sharing of patient information between mutually trusting physicians who are nonetheless not immediately engaged in the treatment of the same individual).

**Definition** **11**(Implicit flow)**.**
*An implicit flow arises without a causal relationship between the entity whose data is subject to the flow and the occurrence of the flow but rather due to a causal relationship between another entity’s decision and the occurrence of the flow or by circumstance. Thus, an implicit flow involving an entity e can be modeled as a random variable o∼Ber(p), where p is independent of the factors influencing e such that:*
(4)$$o=\{\begin{array}{cc} F & w.p.p\\ ⊥ & w.p.1-p \end{array}$$

An example of an implicit flow is the recording of an individual by a security camera in a public space of which the individual was not aware. Implicit flows are sometimes also called *information leaks* and can arise in a number of systems, even those typically considered perfectly secure. For example, a secret ballot that results in an unanimous vote implicitly reveals the preference of all voters. The quantification of information leakage is central to the study of QIF.

As flows are—by definition—pairwise interactions, analyzing many-to-one and one-to-many communication thus requires special consideration. While *one-to-many* communication can be “dismantled” into pairwise flows in a straightforward way, *many-to-one* communication requires considering ownership and governance of the transmitted information. For instance, many-to-one communication where each sender has conjunct governance and ownership of their data can be easily modeled as separate instances of pairwise flows. However, when governance is disjunct or when correlations exist between data, it is required to “marginalize” the contribution of the entities whose data is involved in the flow, even if they themselves are not part of it. Thus, many-to-one-communication can lead to implicit flows arising. This type of phenomenon is an emergent behavior in systems exhibiting complex information flows such as societies and has been described with the term *information bundling problem* by [26]. For example, the message “I am an identical twin” flowing from a sender to a receiver reduces the receiver’s uncertainty about the sender’s sibling’s biological gender and genetic characteristics. As data owned by the sibling and governed by the sender is flowing, information can be considered as implicitly flowing from the sibling to the receiver.

Finally, equipped with the primitives above, we can define *privacy*:

**Definition** **12**(Privacy)**.**
*Let F be a flow of a message M between a sender S and a receiver R over a channel C embedded in a context C. Then,* privacy *is the ability of S to upper-bound the information content of M and of any computation on M, independent of the receiver’s prior knowledge.*

The following implications follow immediately from the aforementioned definition:It relates directly to an *ability* of the sender. We contend that this formulation mirrors the widespread perception of privacy, e.g., as it is formulated in laws. Here, the right to privacy stipulates a legal protection of the ability to restrict information about certain data;Our definition, like our primitives, is atomic. It is possible to maintain privacy *selectively*, i.e., about a single datum. This granularity is required as privacy cannot be viewed “in bulk”;Privacy is contextual. The factors inherent to the specific context in which an information flow occurs (such as trust or incentives above) and the setting of the flow itself therefore largely determine the resulting expectations and entity behaviors, similar to Contextual Integrity. As an example, a courtroom situation (in which the individual is expected to tell the truth and disclose relevant information) is not a privacy violation, as the ability of the individual to withhold information still exists, but the individual may choose to not exercise it. On the flip side, tapping an individual’s telephone *is* a privacy violation, independently of whether it is legally acceptable or illegal. Our framework thus separates between privacy as a faculty and the circumstances under which it is acceptable to maintain it. Edge cases also exist: for example, divulging sensitive information under a threat of bodily harm or mass surveillance states where every privacy violation is considered acceptable would have to be treated with special care (and interdisciplinary discourse) to be able to define what constitutes (or not) a socially acceptable and “appropriate” information flow.

## 4. Connections to PETs

PETs are technologies that aim to offer some quantifiable guarantee of privacy through purely technical means. The fact that the term *privacy* is used loosely harbors considerable risks, as the subject of protection is very often **not** privacy in the sense above. Our framework is naturally suited to analyzing the guarantees provided (or not) by various techniques considered PETs. In the current section, we discuss how DP naturally fulfills our definition, while anonymization techniques do not. Of note, we rely on Shannon’s information theory to quantify the information content in this section.

### 4.1. Anonymization and Its Variants

Anonymization techniques have a long history in the field of private data protection and can be considered the archetypal methods to protect privacy. Except anonymization (i.e., the removal of identifiable names from sensitive datasets), a broad gamut of similar techniques has been proposed, e.g., *k*-anonymity [13]. It is widely perceived among the general population that this offers security against re-identification and hence preserves privacy. However, prior work on *de-anonymization* has shown [27] that anonymization is not resilient to auxiliary information and that the guarantees of techniques like *k*-anonymity degrade unpredictably under post-processing of the message [28]. As described in the definition of privacy above, the information content of the message should not be able to be arbitrarily increased by any computation on it or by any prior knowledge (auxiliary information) the receiver has. Therefore, none of these techniques offer privacy in the sense described above but are solely means to hinder private information from being immediately and plainly readable. This is mirrored by newer legal frameworks like the European General Data Protection Regulation (GDPR), as discussed below.

### 4.2. Differential Privacy

Differential privacy (DP) [11] is a formal framework and collection of techniques aimed at allowing analysts to draw conclusions from datasets while protecting individual privacy. The guarantees DP offers are *exactly* compatible with our definition of privacy, rendering DP the gold-standard technique for privacy protection within a specific set of requirements and settings. To elaborate this correspondence, we provide some additional details on the DP guarantee. We constrain ourselves to the discussion of ε-DP and *local* DP in the current work.

Consider an entity *E* holding data $D^{E}$ (we deviate from our usage of lowercase symbols for entities to avoid confusion with Euler’s number in this section). Assume the entity wants to transmit a message M concerning a datum $X\in D^{E}$. Let *X* be a random variable taking values in X⊂R, where X has cardinality n>0. Consider a flow $F:E\overset{M}{\to }R$, where R is a receiver. Then, preserving privacy regarding *X* under our definition is the ability of *E* to upper-bound the information content of M about *X*.

Assume now that *E* applies a DP *mechanism*, that is, a randomized algorithm A:X→Y operating on *X* to produce a *privatized* output $Y∼p_{A,X}\left(x\right)$, which forms the content of the message M. We omit the subscript on the probability mass function for readability in the following. This procedure forms the following Markov chain:(5)$$X\overset{A}{\to }Y\to M,$$
where M is observed by R. The fact that A preserves (local) ε-DP offers the guarantee that $\forall \;x,x^{\prime }\in X\;|\;g\left(x,x^{\prime }\right)\leq 1$, where *g* is the discrete metric, and ∀y∈Y such that the following holds:(6)$$p\left(A\left(x\right)=y\right)\leq e^{\varepsilon }p\left(A\left(x^{\prime }\right)=y\right).$$
We note that the guarantee is given over the randomness of A. We show that this implies an upper bound of $\log_{2}e^{\varepsilon }$ on the mutual information between *X* and *Y* and therefore on the amount of information *Y* (and thus M) “reveals” about the true value of *X*. In this sense, the application of a DP mechanism is a sufficient measure to upper-bound the information of M.

**Proposition** **1.**
*Let A,X,Y be defined as above. Then, if A satisfies ε-DP, the following holds:*

(7)
$$I\left(Y\|X\right)=I\left(A\left(X\right)\|X\right)\leq \log_{2}e^{\varepsilon }\mathrm{Sh},$$

*where I(·‖·) denotes the mutual information, and Sh is the Shannon unit of information.*


**Proof.** We begin by re-writing Equation (Equation 6) for readability:
(8)$$p\left(A\left(x\right)=y\right)\leq e^{\varepsilon }p\left(A\left(x^{\prime }\right)=y\right)\Rightarrow p\left(y|x\right)\leq e^{\varepsilon }p\left(y|x^{\prime }\right).$$
Multiplying both sides by $p(x^{\prime })$, we obtain:
(9)$$p\left(y|x\right)p\left(x^{\prime }\right)\leq e^{\varepsilon }p\left(y|x^{\prime }\right)p\left(x^{\prime }\right)=p\left(y,x^{\prime }\right).$$
Marginalizing out $x^{\prime }$, we have:
(10)$$\sum_{x^{\prime }}p\left(y|x\right)p\left(x^{\prime }\right)\leq e^{\varepsilon }\sum_{x^{\prime }}p\left(y,x^{\prime }\right)\Rightarrow p\left(y|x\right)\leq e^{\varepsilon }p\left(y\right).$$
The above can be rewritten as:
(11)$$\frac{p(y|x)}{p(y)}=\frac{p(y|x)p(x)}{p(y)p(x)}=\frac{p(x,y)}{p(x)p(y)}\leq e^{\varepsilon }.$$
Taking the logarithm of both sides and applying the expectation operator, we obtain:
(12)$$E_{p_{XY}}\left[\log_{2}\left(\frac{p(x,y)}{p(x)p(y)}\right)\right]\leq E_{p_{XY}}\left[\log_{2}e^{\varepsilon }\right]=\log_{2}e^{\varepsilon }.$$
The left hand side of Equation (Equation 12) is the mutual information I(Y‖X), from which the claim follows. □

**Remark** **6.**
*By the information processing inequality, this also bounds the mutual information between X and M in Equation (Equation 5).*


DP has additional beneficial properties, such as a predictable behavior under composition, whereby *n* applications of the DP mechanism will result in a cumulative information content of at most $\log_{2}e^{n\varepsilon }$. Moreover, like our definition, (ε-)DP holds in the presence of computationally unbounded adversaries and is closed under arbitrary post-processing. This renders DP a powerful and general tool to satisfy our privacy definition in a variety of scenarios. Its utilization is often aimed not at individual data records (such as the local DP example above) but at statistical databases, where it gives guarantees of privacy to every individual in the database. In fact, it is simple to generalize the proposition above to such statistical databases to show that *every* individual in a database enjoys the same guarantee of bounded information about their personal data.

Despite the strong and natural links between our privacy definition and DP, it cannot be claimed that they are identical. For one, DP is a quantitative definition and is not designed to handle semi-quantitative notions of information content such as structural or metric information content. Nonetheless, the similarities between ensuring DP and ensuring privacy in our sense are striking: Whereas DP is a notion of privacy that can be implemented using *statistical noise*, one could think about semantic, metric, or structural privacy as being implemented using *communication noise* [29]. A simple example of such *noise* is transmitting false information, which—under our definition—can be used as a form of privacy preservation. We expressly note that the inverse does not hold: Whereas a flow satisfying DP also satisfies our definition of privacy, satisfying our privacy definition is not a valid DP guarantee. This is also simple to mathematically verify by, e.g., showing that a bound on mutual information does not represent a useful DP guarantee (as it translates to a bound on total variation distance between the input and output distributions, thus not bounding the magnitude of a worst-case event but only its probability of occurrence).

Another point of differentiation between our privacy definition and DP is context-reliance. DP is—by and large—a guarantee that does not concern itself with context. This is a “feature” of DP and not a shortcoming and what renders it powerful and flexible. However, there exist situations in which a valid DP guarantee may not translate directly into an acceptable and relatable result. Consider the example of publishing an image under local DP. Even though the direct addition of noise to an image may satisfy a DP guarantee, the amount of noise that is required to be added to satisfy a guarantee that is considered acceptable by most individuals (that is, one which hides relevant features) is likely to render the image entirely unrecognizable, also nullifying utility. However, most individuals would consider an appropriately blurred image as preserving acceptable privacy, even though such a blurring operation (especially if carried out on only parts of the image) may be difficult to analyze under the DP lens. We nonetheless consider the development of rigorous and quantifiable DP guarantees for such scenarios a promising and important future research direction.

## 5. Discussion

### 5.1. Why Is Privacy Difficult to Protect in Practice?

Our definitional framework sheds light on many of the challenges of protecting privacy *in the real world*. These challenges arise from a discrepancy created by the assumptions required to formally define what privacy is and the facets of human communication. We highlight some of these challenges in this section.

The first fundamental challenge in preserving privacy is the difficulty in **assigning a flow to its origin**. The complexity of this task is two-fold. As discussed under *implicit flows* above, it means determining which *entity* the flow originated from, as communication can often (possibly involuntarily) involve information of more than one entity. Consider the following context C from the example with the identical twins above. The flow $F_{1}$ involves sender $S_{1}$ and receiver R. However, in revealing that they are an identical twin, information starts flowing from $S_{2}$, the sibling to R, thus inducing an additional (implicit) information flow context $C_{2}$.

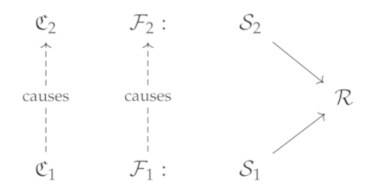
(13)

The second facet of the challenge arises from attempting to resolve the nomic association of an informational representation with its associated datum as well as its strength. Mathematically, this problem is equivalent to exact inference on a causal Bayesian graph. Consider the following causal graphical model:

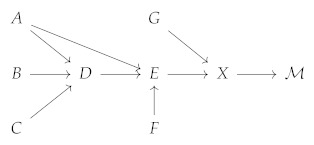
(14)

Here, (A,…,X) are random variables, and we assumed that all arrows indicate causal relationships, that is, the state of the variable at the origin of the arrow causes the state of the variable at its tip. Even in this relatively simple example, and given that the causal relationships are known, *X* contains information about $\left(A,\ldots ,G\right)$. Moreover, there is both a fork phenomenon between A,D, and *E* and a collider phenomenon between G,E, and *X*. The determination of how much information was revealed about each of the variables by transmitting M therefore requires factoring the graph into its conditional probabilities, which is, in general, NP-hard. However, such graphs, and even much more complicated ones, are likely very typical for human communication, which is non-atomic and thus contains information about many different items of the entity’s data. Moreover, determining causality in such settings can be an impossible task. These findings underscore why it is considerably easier to protect privacy in a quantitative sense in statistical databases (where singular items of data are captured) than in general communication and why it is likely impossible to reason quantitatively about privacy in the setting of human communication “in the wild” without making a series of assumptions. The aforementioned difficulties are not unique to the quantitative aspects of our definition but also inherent to DP. The DP framework does not make assumptions about the data of individuals in a database not being correlated with each other; however, the outcome of such a situation may be surprising to individuals who believe their data to be protected when—against their expectation—inferring an attribute the individual considers “private” becomes possible by observing the data of another individual. In general, much like DP, our privacy definition does not consider statistical inference a privacy violation. As an example, consider an individual that participates in a study for a novel obesity medication. Even if the study is conducted DP, it is *not* a privacy violation to determine that the individual is probably overweight. However, learning their exact weight *can* be considered a privacy violation. These examples also highlight the minute, but important, difference between data that is merely *personal* (in this case, the fact that the individual is overweight) and data that is truly *private* (here, their concrete weight). It also motivates a view shared by both our definition and the DP definition, namely, that preserving privacy can be thought of as imposing a bound on the *additional* risk an individual incurs by committing to a transmission of their data. For a formal discussion, we refer to [30].

Last but not least, we emphasize that the very attempt to *measure and restrict* information is a challenging undertaking. In the case of statistical databases and with techniques such as DP, such measurement is possible to an extent, even though certain assumptions may be required. For example, encoding data as a series of yes/no answers may or may not sufficiently represent the true information content of the data but may be required to enable the utilization of a DP technique. In this setting, a choice has to be made between a rigorous method of privacy protection based on Shannon’s information theory and an unedited representation of the entity’s data. The former simplification is often required for machine learning applications (where it is referred to as *feature extraction*). For cases where a rigorous measurement of the message’s information content is not required for a context to acceptably preserve privacy, one of the other aforementioned techniques of restricting information content (e.g., metrically or structurally) may be implemented.

### 5.2. From Formal Privacy to Regulatory Implementation

In this section, we discuss our privacy definition through the lens of legal frameworks by showing that our framework’s primitives exhibit strong connections to the terms stipulated under the European General Data Protection Regulation (GDPR) (https://gdpr.eu/, accessed on 12 May 2022). This (non-exhaustive) example is intended to sketch a possible path towards the adoption of a rigorous definition of privacy by regulators “in the real world”. Moreover, it underscores the interplay of privacy and informational self-determination. Similar parallels could be drawn to, e.g., the California Consumer Privacy Act or related regulations. The GDPR regulates *personal data*, which we refer to as the data owned by an entity. Such data relates to an individual (entity), also called a *data subject*, and it is directly or indirectly identifiable. In our framework, this corresponds to the subset of the entity’s data considered *sensitive*. The GDPR refers to the process we term *governance* above as *data processing*, which is carried out by a *data controller*, corresponding to the entity exercising governance in our framework. Our notion of *delegated governance* is mirrored in the GDPR by the notion of a *data processor*, i.e., a third party processing personal data on behalf of a data controller. We note that distributed governance has no explicit counterpart in the regulation. Finally, we note that pseudonymised (or anonymised) data are *still* considered sensitive under the GDPR if it can be re-identified (i.e., de-pseudo-/anonymised) with little effort. This precisely mirrors our notion that anonymisation and pseudonymisation techniques do not satisfy our privacy definition, as they are partially or fully reversible by the utilization of auxiliary data or other extraneous knowledge and/or post-processing of the message.

We now discuss the relationship of our privacy definition with key parts of the GDPR. We begin by noting that Article 25 of the regulation stipulates that data protection must be implemented *by design and by default*. This means that existent and newly designed systems must consider means to protect personal data a first priority. Our privacy definition’s focus on technical implementation and formal information flow constraints enables system designers to not only implement privacy-enhancing technologies that fulfill this requirement but also quantify their effectiveness when used in practice. Furthermore, Article 6 of the GDPR lists situations in which personal data processing is permissible, e.g., when consent or a legal contract is present. In such cases, our privacy guarantee’s resistance to *post facto* computations on the message signify that—even if the receiving entity violates consent or the contract—no additional information can be gained. This is especially relevant in the case of *legitimate interest*, which is amenable to flexible interpretation by data processors and could be vulnerable to misuse. Last but not least, the rights granted by Chapter 3 of the regulation, and particularly the *right to restrict processing* (Article 18), are enhanced by the inviolable guarantees that technologies like DP offer. However, we remark that two of these rights interact in specific ways with our framework and—by extension—with formal PETs like DP and cryptography. On the one hand, the *right to be forgotten* (Article 17) does not directly fall under the scope of our privacy definition, as we consider it a matter of governance. This categorization is supported by recent findings demonstrating that, although DP and *machine unlearning*, the technical implementation of the right to be forgotten in machine learning algorithms, share certain properties, they have distinct behaviors [31]. On the other hand, the GDPR stipulates restrictions to the rights of Chapter 3 in Article 23. For example, public security may justify a restriction of informational self-determination. Our definition can be applied in two ways in such cases: when entities relinquish their ability to reduce the information of flows originating from them due to trust in regulations such as the GDPR, no privacy violation is observed. When, however, the ability of the individual is subverted (or is forcibly removed as discusses above), a privacy violation takes place. It is then a social matter to decide whether this violation is (legally or ethically) acceptable for the sake of welfare or not. In this sense, our definition views privacy as an ability neutrally and delegates the enactment of the *right to privacy* and the appraisal of its value to society.

## 6. Conclusions and Future Work

We introduced an axiomatic privacy definition based on a flexible, information-theoretic formalism. Our framework has a close and natural relationship to the guarantees offered by PETs such as differential privacy, while allowing one to better interpret the potential shortcomings of techniques such as *k*-anonymity, whose guarantees may degrade in an unpredictable fashion in the presence of auxiliary information. Our definition encompasses not only Shannon’s information theory but can also be used to cover structural or metrical interpretations that are encountered in human perception and communication.

Our formalism exhibits strong links to complex systems research [32] and lends itself to experimental evaluation using agent-based models or reinforcement learning. We intend to implement such models of information flow in society, for example, to investigate economic implications of privacy, in future work. Moreover, we intend to propose a more holistic taxonomy of other PETs, such as cryptographic techniques and distributed computation methods. Lastly, we encourage the utilization of our formalism by social, legal, and communication scientists to find a “common ground” of reasoning about privacy and the guarantees offered by various technologies, for which we provide an initial impulse above. Such standard terminology (data ownership, governance, privacy, etc.) will promote a clear understanding of the promises and shortcomings of such technologies and be paramount for their long-term acceptance, the objective discourse about them on a political and social level, and, ultimately, their broad implementation and adoption.

## Data Availability

Not applicable.

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
