# Peer review of "Privacy: An Axiomatic Approach"

_entropy, 2022, doi:10.3390/e24050714_

Round 1

Reviewer 1 Report

This authors aims to formulate an axiomatic definition of privacy relying on the sociological, legal and philosophical privacy research. The paper is intriguing and has a good grade of novelty, and in some parts is pleasant to read.

That said, i think that authors should make an effort to link their privacy concept with "real-world" privacy (i.e., privacy stated by laws and regulations, such as GDPR). I understand that authors aim to formulate privacy in a "future-proof" manner, resilient to the change of laws and technologies, but from my point of view the link suggested will greatly increase the interest of readers.

Page 2, rows 45-ff. Some authors highlight limits and misuse of Differential Privacy technique. A must-read paper about those issues is:

Domingo-Ferrer, J., Sánchez, D., & Blanco-Justicia, A. (2021). The limits of differential privacy (and its misuse in data release and machine learning). Communications of the ACM64(7), 33-35.

This is a short paper, but its references Section will help to find other sources to go deeper into those issues.

Author Response

That said, i think that authors should make an effort to link their privacy concept with "real-world" privacy (i.e., privacy stated by laws and regulations, such as GDPR). I understand that authors aim to formulate privacy in a "future-proof" manner, resilient to the change of laws and technologies, but from my point of view the link suggested will greatly increase the interest of readers.

We welcome this suggestion by the reviewer and have included a discussion of how our definition relates to the GDPR as a separate subsection in the Discussion. The added text reads as follows:

In this section, we discuss our privacy definition through the lens of legal frameworks by showing that our framework's primitives exhibit strong connections to the terms stipulated under the European General Data Protection Regulation (GDPR). This (non-exhaustive) example is intended to sketch a possible path towards the adoption of a rigorous definition of privacy by regulators ”in the real world”. Moreover, it underscores the interplay of privacy and informational self-determination. Similar parallels could be drawn to e.g. the California Consumer Privacy Act or similar regulations. The GDPR regulates personal data, which we refer to as the data owned by an entity. Such data relates to an individual (entity), also called a data subject and is directly or indirectly identifiable. In our framework, this corresponds to the subset of the entity's data considered sensitive. The GDPR refers to the process we term governance above as data processing, which is carried out by a data controller, corresponding to the entity exercising governance in our framework. Our notion of delegated governance is mirrored in the GDPR by the notion of a data processor, i.e. a third party processing personal data on behalf of a data controller. We note that distributed governance has no explicit counterpart in the regulation. Finally, we note that pseudonymised (or anonymised) data is still considered sensitive under the GDPR if it can be re-identified (i.e. de-pseudo-/anonymised) with little effort. This precisely mirrors our notion that anonymisation and pseudonymisation techniques do not satisfy our privacy definition, as they are partially or fully reversible by the utilisation of auxiliary data or other extraneous knowledge and/or post-processing of the message.

We now discuss the relationship of our privacy definition with key parts of the GDPR. We begin by noting that Article 25 of the regulation stipulates that data protection must be implemented by design and by default. This means that existent and newly designed systems must consider means to protect personal data as a first priority. Our privacy definition's focus on technical implementation and formal information flow constraints enables system designers to not only implement privacy-enhancing technologies which fulfill this requirement, but also quantify their effectiveness when used in practice. Furthermore, Article 6 of the GDPR lists situations in which personal data processing is permissible, e.g. when consent or a legal contract is present. In such cases, our privacy guarantee's resistance to post facto computations on the message signify that --even if the receiving entity violates consent or the contract-- no additional information can be gained. This is especially relevant in the case of so-called legitimate interest, which is amenable to flexible interpretation by data processors and could be vulnerable to misuse. Last but not least, the rights granted by Chapter 3 of the regulation, and particularly the right to restrict processing (Article 18) are enhanced by the inviolable guarantees that technologies like DP offer. However, we remark that two of these rights interact in specific ways with our framework and --by extension-- with formal PETs like DP and cryptography. On one hand, the right to be forgotten (Article 17) does not directly fall under the scope of our privacy definition, as we consider it a matter of governance. This categorisation is supported by recent findings demonstrating that, although DP and machine unlearning, the technical implementation of the right to be forgotten in machine learning algorithms, share certain properties, they have distinct behaviours. On the other hand, the GDPR stipulates restrictions to the rights of Chapter 3 in Article 23. For example, public security may justify a restriction of informational self-determination. Our definition can be applied in two ways in such cases: when entities relinquish their ability to reduce the information of flows originating from them due to trust in regulations such as the GDPR, no privacy violation is observed. When however the ability of the individual is subverted (or is forcibly removed as above), a privacy violation exists. It is then a matter of society to decide whether this violation is (legally or ethically) acceptable for the sake of welfare or not. In this sense, our definition views privacy as an ability neutrally, and delegates the enactment of the right to privacy and the appraisal of its value, to society.

Page 2, rows 45-ff. Some authors highlight limits and misuse of Differential Privacy technique. A must-read paper about those issues is:

Domingo-Ferrer, J., Sánchez, D., & Blanco-Justicia, A. (2021). The limits of differential privacy (and its misuse in data release and machine learning). Communications of the ACM, 64(7), 33-35.

This is a short paper, but its references Section will help to find other sources to go deeper into those issues.

We thank the reviewer and have included the citation in the Prior Work section. The corresponding section reads as follows:

We note that, whereas our discussion focuses on DP, which is rooted in the work of Dwork et al.(Dwork2013), DP is not the only PET, nor the only way to assure that our definition of privacy is fulfilled (...). Despite criticism regarding the guarantees and limitations of DP (Domingo et al.), it has established itself as the gold standard for privacy protection in fields such as statistics on large databases.

Reviewer 2 Report

The paper "Privacy: An Axiomatic Approach" presents an axiomatic definition of privacy based on quantifiable information flows. The manuscript is well-structured, and readable, and the reader can easily follow. However, there are some points that need to be improved in order to increase the quality of the work:

  • In Section 3, the authors state: "for example, divulging sensitive information under a threat of bodily harm or mass surveillance states where every privacy violation is considered acceptable are not within the scope of our definition, which assumes a well-functioning society."

    Perhaps such examples should be considered to be in scope (of any definition); after all, which society today can be defined as a well-functioning one?
  • In Section 6, the authors state: "Our framework […] explains why techniques such as anonymization, which purportedly preserve privacy, in actuality do not"

    It is a rather bold statement to claim that well-established techniques (like anonymization or k-anonymization) do not work. It was not made clear throughout the manuscript why such techniques do not work.

Author Response

In Section 3, the authors state: "for example, divulging sensitive information under a threat of bodily harm or mass surveillance states where every privacy violation is considered acceptable are not within the scope of our definition, which assumes a well-functioning society."

Perhaps such examples should be considered to be in scope (of any definition); after all, which society today can be defined as a well-functioning one?

We thank the reviewer for this remark. We have rephrased the section to incorporate the reviewer’s input, which now reads as follows:

(...) divulging sensitive information under a threat of bodily harm or mass surveillance states where every privacy violation is considered acceptable would have to be treated with special care (and interdisciplinary discourse) to be able to define what constitutes (or not) a socially acceptable and appropriate information flow.

In Section 6, the authors state: "Our framework […] explains why techniques such as anonymization, which purportedly preserve privacy, in actuality do not"

It is a rather bold statement to claim that well-established techniques (like anonymization or k-anonymization) do not work. It was not made clear throughout the manuscript why such techniques do not work.

We thank the reviewer for this remark and agree that the statement is perhaps a bit too drastic. We have rephrased the statement to clarify which guarantees are preserved by techniques like k-anonymity and differential privacy. The corresponding sections now read as follows:

However, prior work on de-anonymization has shown that anonymization is not resilient to auxiliary information and that the guarantees of techniques like k-anonymity degrade unpredictably under post-processing of the message (de Montjoye, 2013). As described in the definition of privacy above, the information content of the message should not be able to be arbitrarily increased by any computation on it or by any prior knowledge (auxiliary information) the receiver has. Therefore, none of these techniques offer privacy in the sense described above, but are solely means to hinder private information from being immediately and plainly readable. This is mirrored by newer legal frameworks like the European General Data Protection Regulation (GDPR), as discussed below.

Our framework has a close and natural relationship to the guarantees offered by PETs such as differential privacy, while allowing one to better interpret the potential shortcomings of techniques such as k-anonymity, whose guarantees may degrade in an unpredictable fashion in the presence of auxiliary information.